# The Elastic Lottery Ticket Hypothesis

**Xiaohan Chen**[1][*]   **Yu Cheng**[2]   **Shuohang Wang**[2]   **Zhe Gan**[2]
**Jingjing Liu**[3]   **Zhangyang Wang**[1]

[1]University of Texas at Austin   [2]Microsoft Corporation   [3]Tsinghua University
{xiaohan.chen, atlaswang}@utexas.edu
{yu.cheng, shuowa, zhe.gan}@microsoft.com
JJLiuiu@air.tsinghua.edu.cn

## Abstract

Lottery Ticket Hypothesis (LTH) raises keen attention to identifying sparse trainable subnetworks, or winning tickets, which can be trained in isolation to achieve similar or even better performance compared to the full models. Despite many efforts being made, the most effective method to identify such winning tickets is still *Iterative Magnitude-based Pruning (IMP)*, which is computationally expensive and has to be run thoroughly for every different network. A natural question that comes in is: *can we "transform" the winning ticket found in one network to another with a different architecture, yielding a winning ticket for the latter at the beginning, without re-doing the expensive IMP?* Answering this question is not only practically relevant for efficient "once-for-all" winning ticket finding, but also theoretically appealing for uncovering inherently scalable sparse patterns in networks. We conduct extensive experiments on CIFAR-10 and ImageNet, and propose a variety of strategies to tweak the winning tickets found from different networks of the same model family (e.g., ResNets). Based on these results, we articulate the *Elastic Lottery Ticket Hypothesis* (**E-LTH**): by mindfully replicating (or dropping) and re-ordering layers for one network, its corresponding winning ticket could be stretched (or squeezed) into a subnetwork for another deeper (or shallower) network from the same family, whose performance is nearly the same competitive as the latter's winning ticket directly found by IMP. We have also extensively compared E-LTH with pruning-at-initialization and dynamic sparse training methods, as well as discussed the generalizability of E-LTH to different model families, layer types, and across datasets. Code is available at `https://github.com/VITA-Group/ElasticLTH`.

## 1   Introduction

*Lottery Ticket Hypothesis (LTH)* [13] suggests the existence of sparse subnetworks in over-parameterized neural networks at their random initialization, early training stage, or pre-trained initialization [35, 44, 5, 3, 2]. Such subnetworks, usually called *winning tickets*, contain much fewer non-zero parameters compared with the original dense networks, but can achieve similar or even better performance when trained in isolation. The discovery undermines the necessity of over-parameterized initialization for successful training and good generalization of neural networks [46, 31]. That implies the new possibility to train a highly compact subnetwork instead of a prohibitively large one without compromising performance, potentially drastically reducing the training cost.

However, the current success of LTH essentially depends on *Iterative Magnitude-based Pruning (IMP)*, which requires repeated cycles of training networks from scratch, pruning and resetting the remaining parameters. IMP makes it extremely expensive and sometimes unstable to find winning

---

[*] Work was done when the author interned at Microsoft.

35th Conference on Neural Information Processing Systems (NeurIPS 2021).

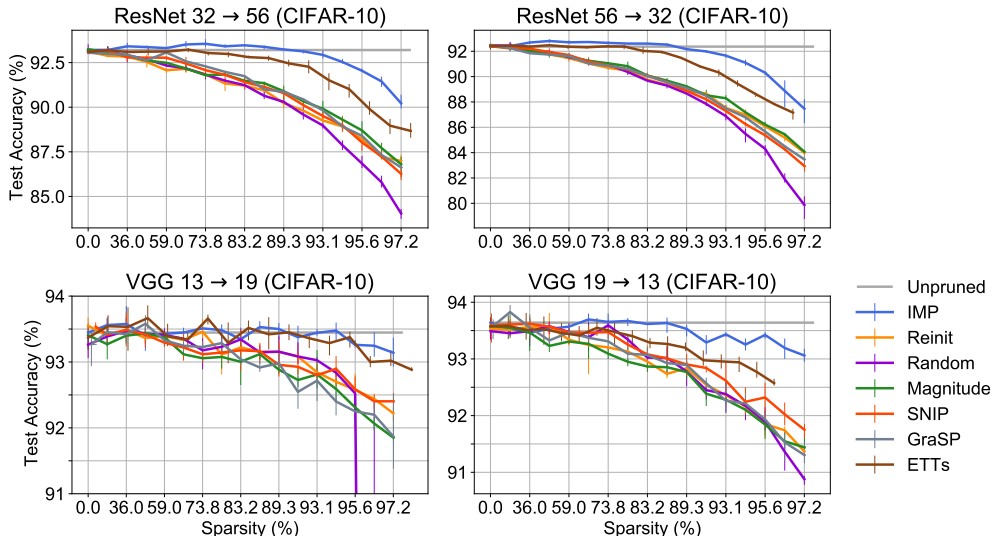

Figure 1: Validating Elastic Lottery Ticket Hypothesis (E-LTH) between two pairs of networks (ResNet-32 and ResNet-56, VGG-13 and VGG-19) trained on the CIFAR-10 dataset. We transform the winning tickets in the source models at different levels of sparsity using the proposed *Elastic Ticket Transformations (ETTs)*, and compare with other pruning methods (please refer to Section 4 for those methods' details), including the state-of-the-art pruning-at-initialization methods (e.g., SNIP and GraSP). Substantial accuracy gaps can be observed between ETTs and other pruning methods, corroborating that ETTs provide high-quality tickets on the target models that are comparable with the winning tickets found by the expensive IMP. All results are based on three independent runs.

tickets at scale, with large models and large datasets [15]. To alleviate this drawback, many efforts have been devoted to finding more efficient alternatives to IMP that can identify sparse trainable subnetworks at random initialization, with little-to-no training [27, 40, 38, 16]. Unfortunately, they all see some performance gap when compared to the winning tickets found by IMP, with often different structural patterns [16]. Hence, IMP remains to be the *de facto* scheme for lottery ticket finding.

Prior work [33] found that a winning ticket of one dense network can generalize across datasets and optimizers, beyond the original training setting where it was identified. Their work provided a new perspective of reducing IMP cost – to only find one generic, dataset-independent winning ticket for each backbone model, then transferring and re-training it on various datasets and downstream tasks. Compared to this relevant prior work which studies the transferablity of a winning ticket in the same network architecture, in this paper, we ask **an even bolder question**:

*Can we transfer the winning ticket found for one network to other different network architectures?*

This question not only has strong practical relevance but also arises theoretical curiosity. On the practicality side, if its answer is affirmative, then we will perform only expensive IMP for one network and then automatically derive winning tickets for others. It would point to a tantalizing possibility of *once-for-all* lottery ticket finding, and the extraordinary cost of IMP on a "source architecture" is amortized by transferring to a range of "target architectures". A promising application is to first find winning tickets by IMP on a small source architecture and then transfer it to a much bigger target architecture, leading to drastic savings compared to directly performing IMP on the latter. Another use case is to compress a larger winning ticket directly to smaller ones, in order to fit in the resource budgets on different platforms. On the methodology side, this new form of transferability would undoubtedly shed new lights on the possible mechanisms underlying the LTH phenomena by providing another perspective for understanding lottery tickets through their transferablity, and identify shared and transferable patterns that make sparse networks trainable [17, 38, 14]. Moreover, many deep networks have regular building blocks and repetitive structures, lending to various model-based interpretations such as dynamical systems or unrolled estimation [19, 42, 20, 1]. Our explored empirical methods seem to remind of those explanations too.

## 1.1 Our Contributions

We take the first step to explore how to transfer winning tickets across different architectures. The goal itself would be too daunting if no constraint is imposed on the architecture differences. Just like

general transfer learning, it is natural to hypothesize that two architectures must share some similarity so that their winning tickets may transfer: *we stick to this assumption in this preliminary study*.

We focus our initial agenda on network architectures from the same design family (e.g., a series of models via repeating or expanding certain building blocks) but of different depths. For a winning ticket found on one network, we propose various strategies to "stretch" it into a winner ticket for a deeper network of the same family, or "squeeze" it for a shallower network. We then compare their performance with tickets directly found on those deeper or shallower networks. We conduct extensive experiments on CIFAR-10 with models from the ResNet and VGG families, and further extend to ImageNet. Our results seem to suggest an affirmative answer to our question in this specific setting.

We formalize our observations by articulating the *Elastic Lottery Ticket Hypothesis* (**E-LTH**): by mindfully replicating (or dropping) and re-ordering layers for one network, its corresponding winning ticket could be stretched (or squeezed) into a subnetwork for another deeper (or shallower) network from the same family, whose performance is nearly the same as the latter's winning ticket directly found by IMP. Those stretched or squeezed winning tickets largely outperform the sparse subnetworks found by pruning-at-initialization approaches [27], and show competitive efficiency to state-of-the-art dynamic sparse training [11]. We also provide intuitive explanations for the preliminary success.

Lastly, we stress that our method has a pilot-study nature, and is not assumption-free. The assumption that different architectures come from one design "family" might look restrictive. However, we point out many state-of-the-art deep models are designed in "families", such as ResNets [22], MobileNets [23], EfficientNets [37], and Transformers [39]. Hence, we see practicality in the current results, and we are also ready to extend them to more general notions - which we discuss in Section 5.

## 2 Related Work

### 2.1 Lottery Ticket Hypothesis Basics

The pioneering work [13] pointed out the existence of winning tickets at random initialization, and showed that these winning tickets can be found by IMP. Denote $f(x; \theta)$ as a deep network parameterized by $\theta$ and $x$ as its input. A sub-network of $f$ can be characterized by a binary mask $m$, which has exactly the same dimension as $\theta$. When applying the mask $m$ to the network, we obtain the sub-network $f(x; \theta \odot m)$, where $\odot$ is the Hadamard product operator.

For a network initialized with $\theta_0$, the IMP algorithm with rewinding [15, 14] works as follows: (1) initialize $m$ as an all-one mask; (2) train $f(x; \theta_0 \odot m)$ for $r$ steps to get $\theta_r$; (3) continue to fully train $f(x; \theta_r \odot m)$ to obtain a well-trained $\theta$; (3) remove a small portion $p \in (0, 1)$ of the remaining weights with the smallest magnitudes from $\theta \odot m$ and update $m$; (5) repeat (3)-(4) until a certain sparsity ratio is achieved. Note that when $r = 0$, the above algorithm reduces to IMP without rewinding [13]. Rewinding is found to be essential for successfully and stably identifying winning tickets in large networks [15, 14]. [44] identified Early-Bird Tickets that contain structured sparsity, which emerge at the early stage of the training process. [33, 3] studied the transferability of winning tickets between datasets; the former focuses on showing one winning ticket to generalize across datasets and optimizers; and the latter investigates LTH in large pre-trained NLP models, and demonstrates the winning ticket transferability across downstream tasks.

### 2.2 Pruning in the Early Training, and Dynamic Sparse Training

A parallel line of works study the pruning of networks at either the initialization or the early stage of training [6], so that the resulting subnetworks can achieve close performance to the dense model when fully trained. *SNIP* [27] proposed to prune weights that are the least salient for the loss in the one-shot manner. *GraSP* [40] further exploited the second-order information of the loss at initialization and prune the weights that affect gradient flows least. [38] unified these methods under a newly proposed invariant measure called *synaptic saliency*, and showed that pruning iteratively instead of in one-shot is essential for avoiding layer collapse. The authors then propose an iterative pruning method called *SynFlow* based on synaptic saliency that requires no access to data.

However, it is observed in [16] that vanilla magnitude-based pruning [21] is as competitive as the above carefully designed pruning methods in most cases, yet all are inferior to IMP by clear margins. The pruning-at-initialization methods such as SNIP and GraSP were suggested to identify no more than a layer-wise pruning ratio configuration. That was because the SNIP/GraSP masks were found to be insensitive to mask shuffling within each layer, while LTH masks were impacted a lot by the same. Those suggest that existing early-pruning methods are not yet ready to replace IMP.

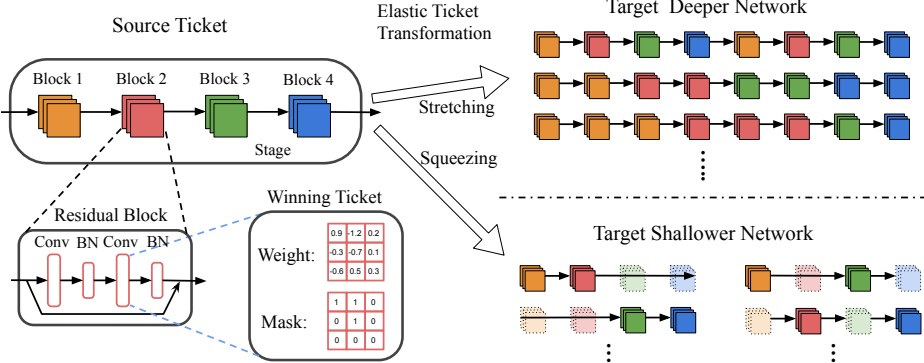

Figure 2: An illustration of Elastic Lottery Ticket Hypothesis, using a ResNet-type network as an example, which includes five residual blocks in each stage. Note that we hide "Block 0" who downsamples the input features as we will always preserve this block, without replicating or dropping. On the left, we use IMP to find winning tickets of the source network. On the right, we can either stretch the source ticket to a deeper network, or squeeze it to a shallower network, using the proposed Elastic Ticket Transformation (ETT). In both cases, we have a number of strategies for mindfully replicating, dropping and re-ordering the blocks.

Instead of searching for static sparse subnetworks, another line of influential works, called *dynamic sparse training (DST)*, focuses on training sparse subnetworks from scratch while dynamically changing the connectivity patterns. DST was first proposed in [32]. Following works improve DST by parameter redistribution [34, 29] and gradient-based methods [8, 11]. A recent work [30] suggested that successful DST needed to explore the training of possible connections sufficiently.

### 2.3 Network Growing

Broadly related to this paper also includes the research on network growing. Net2Net [4] provided growing operations that preserve the functional equivalence for widening and deepening the network. Network Morphism [41] further extended Net2Net to more flexible operations that change the network architecture but maintains its functional representation. Network Morphism is used in [10] to generate several neighbouring networks to select the best one based on some training and evaluation. However, this requires comparing multiple candidate networks simultaneously. A more recent work, FireFly [43], proposed a joint optimization framework that grows the network during training.

Despite our similar goal of transferring across architectures, the above methods do not immediately fit into our case of winning tickets. We face the extra key hurdle that we need to transfer (stretch or squeeze) not only general weights, but also the sparse mask structure while preserving approximately the same sparsity. For example, Net2Net and FireFly add (near) identity layers as the initialization for the extra layers, which are already highly sparse and incompatible with IMP without modification. Network Morphism decomposes one layer into two by alternating optimization, which will break the sparse structure of the source winning tickets.

## 3 Elastic Lottery Ticket Hypothesis

The overall framework of E-LTH is illustrated in Figure 2. At the core of E-LTH are a number of *Elastic Ticket Transformations (ETTs)* proposed by us to convert the sparse architecture of a winning ticket to its deeper or shallower variants. We present rules of thumb for better performance, as well as discussions and intuitive explanation for why they do/do not work. For the simplicity of illustration, we will use the ResNet family as examples in this section.

**Terminology:** Suppose we have identified a winning ticket $f^s(\theta_r^s, m^s)$ on a **source network** using IMP, characterized by the rewinding weights $\theta_r^s$ and the binary mask $m_s$, where $r$ is the rewinding steps and $s$ stands for "source". Our goal is to transform the ticket $f^s$ into a **target network** $f^t\left(g(\theta_r^s, m^s), h(\theta_r^s, m^s)\right)$ with a different depth directly, avoiding running the expensive IMP on the target network again. Here, the superscript $t$ stands for "target". $g(\cdot, \cdot)$ and $h(\cdot, \cdot)$ are the transformation mapping for rewinding weights and the mask, respectively, taking the source weights and source mask as inputs.

## 3.1 Stretching into Deeper Tickets

We first present ETTs for stretching a winning ticket, which select certain layers to replicate, both in the (rewinding) weights and the sparse mask. Below, we discuss the major design choices.

**Minimal unit for replication.** Intuitively, "layer" is the most natural choice for the minimal unit in neural networks. Normalization layers (e.g., Batch Normalization layers [24]) are widely used in modern networks and play essential roles in the successful training of the networks. Therefore, in this paper, ETTs consider a linear or convolutional layer and the normalization layer associated with it as a whole "layer". We find this to work well for VGG networks [36].

However, in ResNet [22] or its variants, which consist of multiple residual blocks, we consider the residual block as the minimal unit, because they are the minimal repeating structure. Moreover, the residual blocks, which are composed of two or three convolutional layers with normalization layers and a shortcut path, can be interpreted from different perspectives. For example, one residual block is interpreted as one time step for a forward Euler discretization in dynamical systems [42], or as one iteration that refines a latent representation estimation [19]. More discussions are in Section 3.3.

**Invariant components.** To ensure generalizability, we prefer minimal modification to the architectures involved. Hence, we will keep several components of the networks unaffected during stretching:

- **The number of stages**, which are delimited by the occurrences of down-sampling blocks. Adding new stages change the dimensions of intermediate features and create new layers with totally different dimensions, which are hard to be transferred from source tickets.
- **The down-sampling blocks**. In ResNet networks, the first block of each stage is a down-sampling block. In ETTs, we directly transplant the down-sampling blocks to the target network without modification and keep the one-to-one relationship with the stages.
- **The input and output layers**. Thanks to the first invariance, the input and output layers have consistent dimensions in the source and target networks and thus can be directly reused.

**Which units to replicate?** Let us take ResNet-20 (source) and ResNet-32 (target) as an example. Each stage of ResNet-20 contains one down-sampling block, $B_0^s$, and two normal blocks, $B_1^s$ and $B_2^s$, while ResNet-32 has four normal blocks for each stage. To stretch a winning ticket of ResNet-20 to ResNet-32, we need to add two residual blocks in each stage. Then, *should we replicate both $B_1^s$ and $B_2^s$ once, or only replicate $B_1^s$ (or $B_2^s$) twice? If the latter, which one?*

For the first question, in ETTs we choose to replicate more unique source blocks and for fewer times, i.e., we prefer to replicate both $B_1^s$ and $B_2^s$ once in the above example. As will be explained in Section 3.3, this could be understood as a uniform interpolation of the discretization steps in the dynamical system, a natural way to go finer-resolution along time. Another practical motivation for the choice is that such strategy can better preserve the sparsity ratio of each stage and thus the overall network sparsity. If $B_1^s$ and $B_2^s$ have different sparsity ratios (we observe that usually later blocks are sparser), replicating only one of them for several times will either increase or decrease the overall sparsity; in comparison, replicating both $B_1^s$ and $B_2^s$ proportionally maintains the overall sparsity.

If we compare replicating either $B_1^s$ or $B_2^s$, the former will lead to better performance due to resultant lower sparsity (see above explained), and the latter causing lower performance due to effective higher sparsity. Both are understandable and could be viewed as trade-off options.

**How to order the replicated units?** Following the ResNet-20 to ResNet-32 example above, we replicate $B_1^s$ and $B_2^s$ once for each of them, resulting in two possible ordering of the replicated blocks: (1) $B_0^s \rightarrow B_1^s \rightarrow B_2^s \rightarrow B_1^s \rightarrow B_2^s$, which we call *appending* as the replicated blocks are appended as a whole after the last replicated block; (2) $B_0^s \rightarrow B_1^s \rightarrow B_1^s \rightarrow B_2^s \rightarrow B_2^s$, which we call *interpolation* as each replicated block is inserted right after its source block. We conduct experiments using both ordering strategies and observe comparable performance.

## 3.2 Squeezing into Shallower Tickets

Squeezing the winning tickets in a source network into a shallower network is the reverse process of stretching, and now we need to decide which units to drop. Therefore, we have symmetric design choices as ticket stretching, besides following the same minimal unit and invariances.

To squeeze a winning ticket from ResNet-32 into ResNet-20, we need to drop two blocks in each stage. The first question is: should we drop consecutive blocks, e.g., $B_1^s, B_2^s$ or $B_3^s, B_4^s$, which corresponds to the inverse process of the *appending* ordering, or drop non-consecutive blocks, e.g.,

$B_1^s, B_3^s$ or $B_2^s, B_4^s$, which corresponds to the inverse process of the *interpolation* ordering above? The second question is: in either case, should we drop the earlier or the later blocks?

According to the ablation study in Section 4.1, ETTs are not sensitive to whether we drop blocks consecutively or at intervals; however, it is critical that we do not drop too many early blocks.

### 3.3   Rationale and Preliminary Hypotheses

We draw two perspectives that may intuitively explain the effectiveness of ETTs during stretching and squeezing. Note that both explanations are fairly restricted to the ResNet family, while E-LTH seems to generalize well beyond ResNets. Hence they are only our preliminary hypotheses, and further theoretical understandings of ETTs will be future work.

**Dynamical systems perspective:** ResNets have been interpreted as a discretization of dynamical systems [42, 20, 1]. Each residual block in the network can be seen as one step of a forward Euler discretization, with an implicit time step size of an initial value ordinary differential equation (ODE). Under this interpretation, adding a residual block right after each source block, that copies the weights and batch normalization parameters from the source block, can be seen as a uniform interpolation of this forward Euler discretization, by doubling the number of time steps while halving the implicit step size. Under the same unified view, if we replicate only $B_1^s$ (or $B_2^s$), that could be seen as performing non-uniform interpolation, that super-solves only one time interval but not others. Without pre-assuming which time step is more critical, the uniform interpolation is the plausible choice, hence providing another possible understanding of our design choice in Section 3.1.

**Unrolled estimation perspective:** [19] interprets ResNets as unrolled iterative estimation process: each stage has a latent representation for which the first block in this stage generates a rough estimation and the remaining blocks keep refining it. This perspective provides a strong motivation to keep the first block of each stage untouched, as it contributes to the important initial estimate for the latent representation. Replicating or dropping the remaining residual blocks will incrementally affect the latent representation estimation. The interpolation method for stretching then corresponds to running every refining step for multiple times and the appending method corresponds to re-running (part of) the refining process again for better estimation. It is also implied by [19] that since each residual block is incremental, removing blocks only has a mild effect on the final representation, providing intuition for the effectiveness of squeezing tickets by dropping blocks.

## 4   Numerical Evaluations

We conduct extensive experiments on CIFAR-10 [26] and then ImageNet [7], transferring the winning tickets across multiple models from ResNet family and VGG family. The implementation and experiment details are presented in Appendix. We run all experiments three times independently with different seeds. Besides IMP and the unpruned dense models, we will also compare ETTs with state-of-the-art pruning-at-initialization methods, including **SNIP** [27], **GraSP** [40] and One-shot **Magnitude**-based pruning which was suggested by [16] as a strong baseline. We also include two common pruning baselines in LTH works: (1) *Reinitialization* (**Reinit**), which preserves the identified sparse mask but reinitialize the rewinding weights; and (2) **Random** Pruning, which keeps the rewinding weights yet permuting the masks, only preserving the layerwise sparsity ratios.

Throughout this paper, we use "sparsity" or "sparsity ratio" to denote the portion of zero elements in networks due to pruning. Thus, the higher the sparsity ratio, the more parameters pruned. Note that ETTs may cause (slightly) misaligned sparsity ratios with IMP and other pruning methods because replicating and dropping blocks may result in the sparsity changes in a less controllable way. For a fair comparison, all pruning methods will match the sparsity ratios of the winning tickets generated by IMP, to the best possible extent. We apply pruning methods to the rewinding weights instead of initialization because the former was found to substantially improve LTH performance [16].

### 4.1   A Thorough Ablation Study on ResNet-32 and Resnet-56

In this subsection, we conduct an ablation study about the rules of thumb for constructing better ETTs as discussed in Section 3. We investigate the selection of replicated or squeezed units when we apply ETTs to stretch or squeeze a winning ticket, and the order of replicated units for the stretching case. All results reported are the average of three independent runs of experiments.

**Do we really find winning tickets?** To show that the new network generated by ETTs is a winning ticket, we compare the performance of ETTs with the following baselines when transforming ResNet-

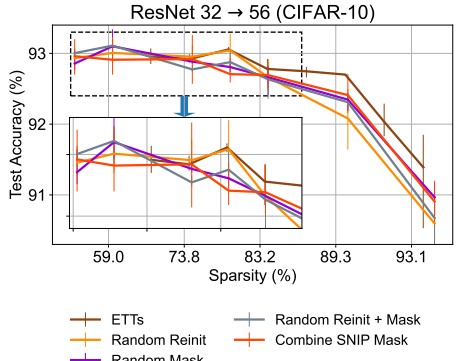
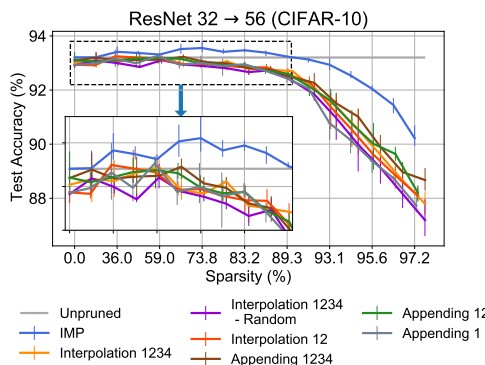

Figure 3: Ablation study to show that ETTs do find winning tickets on the target networks compared to other baselines.

Figure 4: Ablation study on replicating unique units on CIFAR-10, using ETTs to transform tickets in ResNet-32 to ResNet-56.

32 winning tickets into ResNet-56: (1) we transform the mask of the source winning ticket but perform weight reinitialization under that mask; (2) we preserve the weight initialization of the source network but select a random mask; and (3) we use random weights with random mask on the target network. The results are shown in Figure 3, where we can see ETTs are among the best at all non-trivial sparsity levels ($\geq 50\%$) and consistently better than other baselines at high sparsity ratios ($> 80\%$). We also include another baseline which copies the masks from the source winning ticket for existing blocks but uses SNIP masks for the extra blocks in ResNet-56. The superiority of ETTs over this baseline shows that only using the original winning tickets without ETTs is not good enough, corroborating the necessity of ETTs from another perspective.

**Should we replicate more unique units?** ResNet-32 has five residual blocks in each stage, notated as $[B_0, \ldots B_4]$. As we discussed in Section 3, we leave the downsampling block $B_0$ alone and only play with the rest four. We consider three options, i.e., replicating (1) all four blocks; (2) the first two blocks; (3) and the first block ($B_1$). We run for each option both the appending and the interpolation methods (note that appending interpolating the first block yields the same resulting model).

The results are shown in Figure 4, where the numbers in the legend are the indices of replicated blocks. We can see that replicating four blocks is slightly better than replicating two for both appending and interpolation and much better than only replicating the first by clear gap. The gaps enlarge when the sparsity ratios grow higher. At lower sparsity ratios, the differences between those options are smaller. We also include an option, "Interpolation 1234 - Random", in which we randomly permute the masks of the replicated blocks (but preserves the masks of the original blocks). That shows better than its random-pruning variant at all sparsity ratios and the advantage is more evident at high sparsity ratios.

**The earlier or the later units?** The next question in ETTs that follows is: should we replicate earlier units in stretching for better performance, or the later ones? And similarly, should we drop earlier units when squeezing the tickets? Is one option always better than the other? For the stretching part, we try to replicate $(B_1, B_2), (B_2, B_3), (B_3, B_4)$ using appending and interpolation methods. Results in the top subfigure of Figure 5 show the advantage of replicating earlier blocks than the later ones with high sparsity, while the differences are less obvious in the low sparsity range. For squeezing, we also try different options of dropping blocks as shown in the bottom subfigure of Figure 5 and find that ETTs are not sensitive to dropping consecutive or non-consecutive blocks, which is consistent to our observation on the stretching experiments, as long as we do not drop earlier blocks too much – we can see a significant decrease in performance when we drop the first four blocks at the same time.

## 4.2 More Experiments on CIFAR-10

**Experiments on more ResNets.** In addition to the transformation between ResNet-32 and ResNet-56, we apply ETTs that use *appending* operations to more ResNet networks to transfer the winning tickets across different structures. More complete results are presented in Figure 6. We can see that the tickets generated by ETTs from different source networks clearly outperform other pruning methods, with the only exception of transferring then smallest ResNet-14 to large target networks.

Another (perhaps not so surprising) observation we can draw from Figure 6 is that ETTs usually work better when the source and target networks have a smaller difference in depth. For example, when ResNet-44 is the target network, ETT tickets transformed from ResNet-32/56 outperform

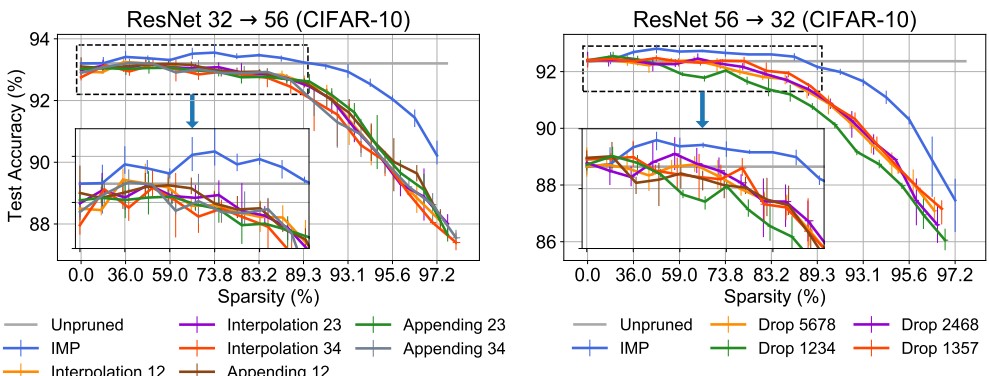

Figure 5: Results of the ablation study on replicating/dropping earlier or later units of the networks. Left: stretching the winning tickets of ResNet-32 into ResNet-56 by appending/interpolating different residual blocks. Right: squeezing the tickets of ResNet-56 into ResNet-32 by dropping four blocks at different positions. Results are the average of three independent runs with error bars.

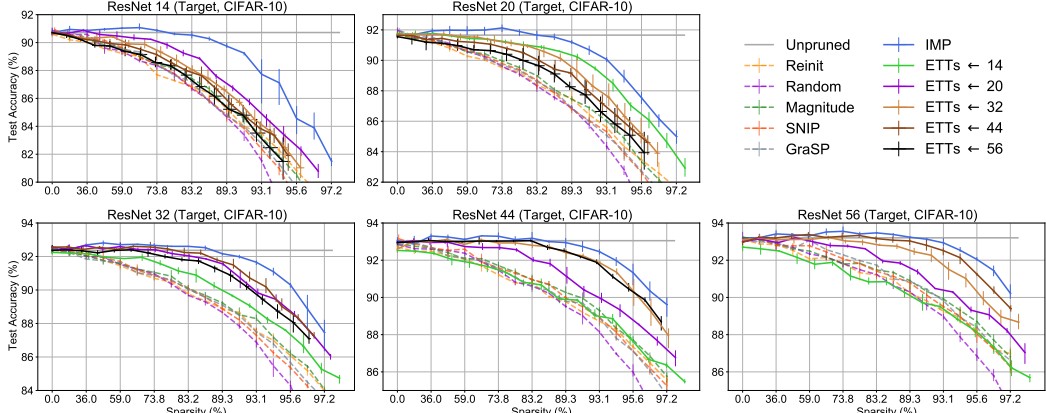

Figure 6: Results of the experiments on CIFAR-10 using 5 different ResNet networks: ResNet-14, ResNet-20, ResNet-32, ResNet-44, ResNet-56, left to right, top to bottom. ETTs $\leftarrow n$ in the legend stands for the ticket transformed from ResNet-$n$ using ETTs, which is also absent in the subfigure where ResNet-$n$ is the target network. Curves for pruning methods are dashed.

those transformed from ResNet-14/20 by notable gaps. Similar comparisons can also be drawn on other target networks. On the other hand, the tickets transformed from ResNet-14 have the best performance than other source networks on ResNet-20, and are nearly as competitive as IMP. However, on ResNet-32, the performance of ETTs from ResNet-14 lies between ETTs from other source models and pruning methods. When the target model goes to ResNet-44 or beyond, ETTs from ResNet-14 perform no better than pruning-at-initialization methods. This implies that our replicating or dropping strategies potentially still introduce noise, which may be acceptable within a moderate range, but might become too amplified when replicating or dropping too many times.

**Experiments on VGG networks.** We also apply ETTs to VGG networks, and report another comprehensive suite of experiments of transforming across VGG-13, VGG-19 and VGG-19 networks. Here we directly replicate/drop "layers", instead of residual blocks. The results shown in Figure 7 convey similar messages as the ResNet experiments, where we see very competitive performance with IMP, even at high sparsity ratios, and significant gaps when compared against other pruning counterparts, especially SNIP and GraSP.

### 4.3 Experiments on ImageNet

Next, we extend the experiments to ImageNet. We adopt three models: ResNet-18, ResNet-26, and ResNet-34. We select sparsity ratio 73.79% as the test bed, at which level IMP with rewinding can successfully identify the winning tickets [15]. Our results are presented in Table 1, where we can see ETTs work effectively when transforming tickets to ResNet-26 and ResNet-34, yielding accuracies comparable to the full model and the IMP-found winning ticket, which are much higher than other

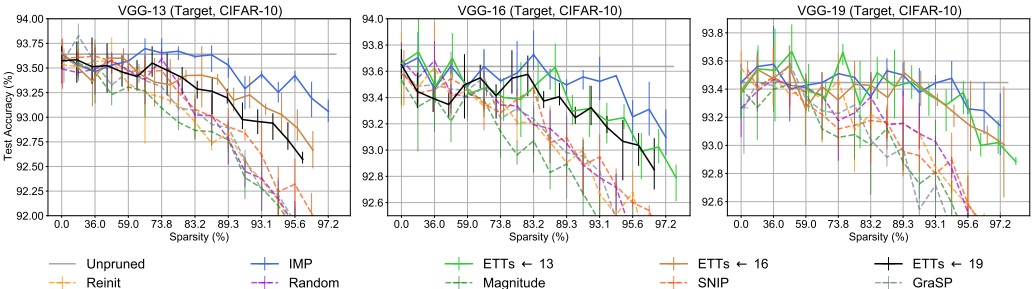

Figure 7: Results of the experiments on CIFAR-10 with VGG networks - VGG-13, -16 and -19. ETTs ← $n$ in the legend stands for the ticket transformed from VGG-$n$ using ETTs, which is also absent in the subfigure where VGG-$n$ is the target network. Curves for pruning methods are dashed.

pruning counterparts. Transforming to ResNet-18 seems slightly more challenging for ETTs, but its superiority over other pruning methods is still solid.

Table 1: E-LTH for ResNets on ImageNet.

| Network | Res-18 | Res-26 | Res-34 |
|---|---|---|---|
| Full Model | 69.96% | 72.56% | 73.77% |
| IMP | 70.22% | 72.94% | 74.20% |
| Reinit | 62.88% | 66.97% | 68.36% |
| Random | 63.10% | 67.07% | 68.68% |
| Magnitude | 64.96% | 68.51% | 69.74% |
| SNIP | 62.23% | 67.51% | 69.35% |
| GraSP | 62.85% | 67.24% | 69.23% |
| ETTs ← 18 | - | 71.29% | 71.86% |
| ETTs ← 26 | **68.17%** | - | **73.37%** |
| ETTs ← 34 | 68.08% | **72.47%** | - |

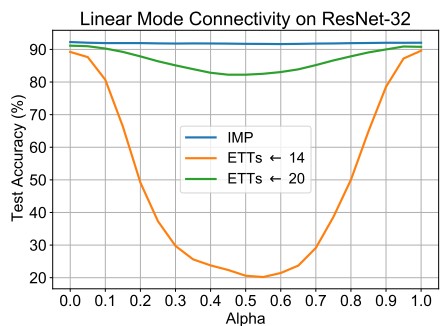

Figure 8: Linear mode connectivity property on ResNet-32. We compare winning tickets directly found on ResNet-32 using IMP and E-LTH tickets transferred from on ResNet-l4 and -20.

## 4.4 Linear Mode Connectivity of E-LTH

[14] observed that winning tickets are stably optimized into a linearly connected minimum under different samples of SGD noises (e.g., data order and augmentation), which means that linear interpolations of two independently trained winning tickets have comparable performance. In contrast, linear interpolations for other sparse subnetworks have severely degraded performance. Here, we observe such "linear mode connectivity" property on tickets transferred from ResNet-14 and ResNet-20 to ResNet-32, shown in Figure 8. We can see that IMP is stable at the linear interpolations of two trained IMP tickets, as observed in [14]. ETTs from ResNet-20 are less stable than IMP in terms of linear mode connectivity, with a maximal 8.72% accuracy drop, but much more stable than ETTs from ResNet-14, with a maximal 69.72% drop. This is consistent with our observation on model accuracies – ETTs have better linear mode connectivity when the source and target networks have a smaller difference in depth, thus having better performance.

**Implication: Possibility to Transfer General Pruned Solutions.** The above experiments indicate lottery tickets transfer better via ETTs when they stay in the same basin, which was found to be the same basin as the pruning solution [12]. That implies ETTs might be able to transfer pruned solutions in general [35] – this is out of the current work's focus, but would definitely be our future work.

## 4.5 Comparison to State-of-the-Art Dynamic Sparse Training Methods

Dynamic sparse training (DST) is an uprising direction to train sparse networks from scratch by dynamically changing the connectivity patterns, while keeping the overall sparse ratios (thus computation FLOPs) low. However, **a crucial difference** between E-LTH and DST is that E-LTH is a "one-for-all" method. For example, if we run IMP on ResNet-34 once, we then obtain tickets for ResNet-20, ResNet-44 and more "for free" simultaneously, by applying ETTs. In contrast, DST methods have to run on each architecture independently from scratch. Therefore, any overhead of E-LTH is *amortized* by transferring to many different-depth architectures: that is conceptually similar to LTH's value in pre-training [2, 3, 18]. E-LTH hence has strong potential in scenarios where we

Table 2: Comparing E-LTH and RigL-5x on ImageNet using ResNet-18/26/34 (use ResNet-26 as the source model).

| Method | E-LTH | RigL-5x |
|---|---|---|
| ResNet-18 | 68.17% | 69.59% |
| ResNet-26 | 72.94% | 72.66% |
| ResNet-34 | 73.37% | 73.88% |

Table 3: Comparing E-LTH with DPF [28] using ResNet-20/32/56 on CIFAR-10, using ResNet-14/20/32 as the source models respectively.

| Backbone | Dense Acc. | Sparsity | Accuracy Drop | |
|---|---|---|---|---|
| | | | DPF | E-LTH |
| ResNet-20 | 92.48% | 90% | -1.60% | -1.64% |
| | | 95% | -4.47% | -4.48% |
| ResNet-32 | 93.83% | 90% | -1.41% | -0.96% |
| | | 95% | -2.89% | -2.83% |
| ResNet-56 | 94.51% | 90% | -0.56% | -0.50% |
| | | 95% | -1.77% | -1.66% |

have different hardware constraints on various types of devices - we can flexibly adapt one ticket to satisfy different constraints without applying IMP or DST methods repeatedly.

To quantitatively understand our relative costs, we compare E-LTH with two recent state-of-the-art DST algorithms, RigL [11] and DPF [28]. To compare E-LTH with RigL, we use ResNet-18/26/34 on ImageNet (ResNet-26 as source model). We set the sparsity ratio to 73.79% so that it is consistent with Section 4.3. For result clarity, we have normalized all FLOPs numbers, by the FLOPs of one-pass standard dense training on ResNet-34[2]. Table 2 indicates comparable accuracies between E-LTH and RigL on three models. RigL uses 0.69x normalized FLOPs to train ResNet-18, 1.0x FLOPs for ResNet-26, and 1.30x FLOPs for ResNet-34. Meanwhile, E-LTH first uses 2.77x FLOPs to find the mask once on ResNet-26; then after applying the (transferred) mask, it only takes 0.14x/0.20x/0.26x additional FLOPs to train ResNet-18/26/34, respectively. Taking together, E-LTH and RigL use similar total FLOPs on training the three models; but note that E-LTH will become more cost-effective as the found mask is transferred to more networks.

To compare E-LTH with DPF, we conduct experiments on CIFAR-10 to compare the accuracies and the FLOPs needed to obtain sparse networks with 90% and 95% sparsity levels. On one hand, we can see from the results in Table 3 that E-LTH achieves accuracy results fully on par with DPF, and even better at higher sparsities of ResNet-32 and ResNet-56. On the other hand, to train sparse models with sparsity ratios of 90% and 95% on all five networks of ResNet-14, -20, -32, -44, and -56, E-LTH needs a total of 1.26 billion FLOPs[3] while DPF requires 1.49 billion. Note that DPF is a dynamic training method and calculates the gradients for all weights (including zero weights), which increases their FLOPs cost; in contrast, E-LTH utilizes static sparsity patterns and thus saves computations during back-propagation by skipping the calculation of the gradients of zero weights.

## 5 Conclusions and Discussions of Broader Impact

This paper presents the first study of the winning ticket transferability between different network architectures (within the same design family), and concludes with the Elastic Lottery Ticket Hypothesis (E-LTH). E-LTH has significantly outperformed SOTA pruning-at-initialization methods, e.g., SNIP and GraSP. Our finding reveals inherent connections between lottery ticket subnetworks derived from similar models, and suggests brand-new opportunities in practice, such as efficient ticket finding for large networks or adaptation under resource constraints. For further discussions about the limitation, deeper questions and future directions, please see the Appendix. We do not see that this work will directly impose any negative social risk. Besides the intellectual merits, the largest potential societal impact that we can see in this work is helping understand lottery tickets and sparse network training. It also provides a solution to flexibly deploy networks with different complexity,

## Acknowledgment

Z.W. is in part supported by the NSF AI Institute for Foundations of Machine Learning (IFML).

---

[2]Different from what was calculated in the **appendix** of [11], here we use a **different sparsity ratio** (73.79%) than the original RigL paper (90%) for fair comparison, and uses the default 5× training steps for the best of RigL performance, which results in (1-73.79%)x5 = 1.30x FLOPs. Similar calculation applies to ResNet-18 and -26. In comparison, E-LTH uses normal training steps (1×).

[3]We run 13 iterations of IMP on ResNet-32 to generate winning tickets with 90% and 95% sparsity ratios and transfer these two tickets to the other four architectures using ETTs.

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
