## A    Experiment Settings

The hyperparameters for the standard training and LTH experiments are shown in Table 4. We follow the official implementation[4] and the hyperparameters of LTH [13, 15]. Rewinding is used by default for IMP.

Table 4: Hyperparameters for network training and IMP, following the settings used in [13, 15]. We conduct all CIFAR-10 experiments for three independent runs with different seeds and one run for ImageNet experiments.

| Network | Dataset | Epochs | Batch | Rate | Schedule | Warmup | Rewinding |
|---------|---------|--------|-------|------|----------|--------|-----------|
| ResNet | CIFAR-10 | 160 | 128 | SGD, 0.1 | x0.1 at 80, 160 epoch | - | 1,000 steps |
|        | ImageNet | 90 | 1,024 | SGD, 0.4 | x0.1 at 30, 60, 80 epoch | 5 epochs | 5 epochs |
| VGG | CIFAR-10 | 160 | 128 | SGD, 0.1 | x0.1 at 80, 160 epoch | - | 200 steps |

## B    Training Time Comparison with RigL

we report a set of training time numbers of E-LTH experiments in Section 4.5 and compare them with RigL. To train a ResNet-18 on ImageNet following the standard 90-epoch training schedule takes 9.0 hours on 8 Nvidia-V100 GPUs, and 10.0 hours and 10.8 hours for ResNet-26 and ResNet-34 respectively. In Section 4.5, to train sparse models of ResNet-18, -26 and -34 at 73.79% sparsity ratio, E-LTH first runs 7 iterations of IMP on ResNet-26, and then trains the transformed LTs of ResNet-18 and ResNet-34 once, inducing a total (7 * 10.0 + 9.0 + 10.8) = 89.8 hours of training. In contrast, RigL independently trains 3 networks with 5x training steps (default setting in the original RigL paper), inducing a total (9.0 + 10.0 + 10.8) * 5 = 149 hours of training.

## C    E-LTH on Fully-Connected Layers

While the above results show the efficacy of E-LTH on convolutional neural networks, the feasibility of applying E-LTH to fully connected (FC) layers remains elusive because replicating FC layers could possibly create dead neurons. Here, we run two sets of experiments to verify so.

In the first setting, we vary the number of FC layers on top of the convolutional layers in VGG-13. We transfer between VGG-13 with 2, 3 (the default option for VGG networks) and 5 FC layers, denoted by VGG-13-2/3/5. In the second setting, we transfer between MLP (multilayer perceptron) models trained on MNIST, characterized by layer width configurations – MLP-n represents a MLP with layer widths $\{784, 300 \times n, 100, 10\}$, where 784 and 10 correspond to the input and output layer, respectively. All experiments are run with 89.26% sparsity ratio. The results of these two settings are shown in Table 5, where E-LTH yields less than 0.21% accuracy drop in all cases, showing that E-LTH also succeeds on FC layers.

## D    Transfer Across Both Architectures and Datasets

Previous to this work, the transferability of lottery tickets was studied in [33, 3]. Here, we also evaluate E-LTH when there are dataset shifts to tentatively investigate the interaction between architecture and dataset transfer. We follow the settings in [33] about dataset transfer. When the source and target model are different, ETTs (from ResNet-20 to ResNet-32) is applied. When only the source and target datasets are different, we directly transfer the winning tickets found by IMP. The baselines accuracies are from the winning tickets directly found on the target model and dataset using IMP. Results are shown in Table 6, where we can see that either transferring across dataset or architecture only induces performance drop, especially when transferring from simple SVHN to more difficult CIFAR-10. But the drops do not stack when we transfer both simultaneously. For example, compared with the original IMP, transferring ResNet-20 ticket on SVHN to ResNet-32 on SVHN yields 0.32% lower accuracy; transferring ResNet-20 ticket from SVHN to CIFAR-10 yields 3.86% accuracy drop.

---

[4]`https://github.com/facebookresearch/open_lth`

Table 5: Results of apply E-LTH to FC layers in VGG networks trained on CIFAR-10 and MLP models trained on MNIST. The sparsity ratio is 89.26% for all experiments.

| Source | Target | | |
|---|---|---|---|
| | VGG-13-2 | VGG-13-3 | VGG-13-5 |
| VGG-13-2 | 93.50% | 93.61% | 93.38% |
| VGG-13-3 | 93.50% | 93.62% | 93.59% |
| VGG-13-5 | 93.44% | 93.59% | 93.38% |
| | MLP-2 | MLP-3 | MLP-4 |
| MLP-2 | 98.06% | 97.95% | 97.99% |
| MLP-3 | 97.91% | 97.83% | 97.83% |
| MLP-4 | 97.88% | 97.91% | 98.03% |

Table 6: Results of dataset transferring between SVHN and CIFAR-10 datasets. When the source and target model are different, ETTs (from ResNet-20 to ResNet-32) is applied. When only the source and target datasets are different, we directly transfer the winning tickets found by IMP.

| Source | Target | | Accuracy | Baseline (IMP) |
|---|---|---|---|---|
| | Model | Dataset | | |
| RN-20 SVHN | SVHN | RN-20 | 95.95% | 95.95% |
| | | RN-32 | 96.00% | 96.32% |
| | CIFAR10 | RN-20 | 87.21% | 91.07% |
| | | RN-32 | 88.67% | 92.22% |
| RN-20 CIFAR10 | SVHN | RN-20 | 95.44% | 95.95% |
| | | RN-32 | 95.79% | 96.32% |
| | CIFAR10 | RN-20 | 91.07% | 91.07% |
| | | RN-32 | 91.38% | 92.22% |

However, transferring from SVHN to CIFAR-10 AND from ResNet-20 to ResNet-32 simultaneously yields only 3.55% drop.

We also successfully transfer winning tickets found on ImageNet to CIFAR-10. We use the winning tickets of ResNet-18 and ResNet-34 found in the ImageNet experiments in Section 4.3 with sparsity ratio 73.79%. When transferring to CIFAR-10, we upsample the CIFAR-10 images to the size of ImageNet images so the samples are compatible with models trained on ImageNet. From the results presented in Table 7, we can see that directly transferring the winning ticket of ResNet-18 reduces almost no accuracy drop. In contrast, transferring the elastic ticket transformed from ResNet-18 to ResNet-26 has slightly worse accuracy but still comparable to the dense model.

Table 7: Results of transferring E-LTH from ImageNet to CIFAR-10.

| Model | Top-1 Acc |
|---|---|
| Dense Res-18 | 95.21% |
| IMP | 95.42% |
| ETTs ← 18 | 95.40% |
| ETTs ← 26 | 95.16% |

## E  Discussions

Despite the existing preliminary findings, there is undoubtedly a huge room for E-LTH to improve. As its foremost limitation at present, the current E-LTH only supports a ticket to scale along the depth dimension. We also make preliminary attempts to extend ETTs to width transformation, but find stretching or squeezing network widths to be much more challenging than depth transformations (see Appendix). We conjecture that width-oriented ETTs will need be inspired and derived from a very different set of tools, than the current depth-oriented tools inspired by Section 3.3. We conjecture some promising new tools can be drawn from the study of ultra-wide deep networks [25].

Even further, what is the prospect for E-LTH to go beyond deepening or widening a source ticket (or vice versa)? Can more sophisticated network topology transforms be considered? Answering this question requires a deep reflection on what makes two architectures "similar", i.e., belong to some same design family with transferable patterns. A recent work [45] shows that many neural network types can be represented by relational graphs with ease, and a graph generator can yield many diverse architectures sharing certain graph measure property (Erdos-Renyi, small-world, etc.). As future work, we plan to leverage their graph generator to systematically explore a design space of neural networks, and see whether there exists elastic winning tickets among many or all of them. We will also continue exploring the theoretical underpinnings of E-LTH.

**Channel pruning or structured sparsity?** Most state-of-the-art LTH and pruning-at-initialization (SNIP, GRASP, SynFlow etc.) algorithms use unstructured sparsity and IMP [13]. [44] tried to identify winning tickets with structured sparsity by channel pruning but its focus is on energy-efficient training and unfortunately sacrifices a lot more accuracy than typical LTH definitions. Therefore, to our best knowledge, finding a high-quality lottery ticket with structured sparsity remains to be an open and unsolved question, and is beyond the subject of this paper. Our work is aimed at improving LTH in terms of transferability instead of developing a general pruning method, and hence we choose to follow the well-accepted standard of this LTH field, by using unstructured sparsity and IMP. Besides, while structured sparsity is more friendly to GPU acceleration, the recent development in hardware accelerators shows the promise to turn the unstructured sparsity into practical speedups. For example, in the range of 70%-90% unstructured sparsity, XNNPACK [9] has already shown significant speedups over dense baselines on real smartphone processors. Those recent advances are likely to mitigate the gap between the current LTH practice and the hardware practice.