# OpenReview forum: "The Elastic Lottery Ticket Hypothesis"
_NeurIPS.cc/2021/Conference — NeurIPS 2021 Poster_

### Official Review · Reviewer_GtXb · 2021-07-01

**Rating:** 6
**Confidence:** 5

**Summary:**

This paper transfer the winning ticket found for one model to other models, the author claims it can reduce the pruning cost for a large-scale model. The motivation is clear.




**Limitations And Societal Impact:**

Yes

**Main Review:**

However, I still have some concerns about this manuscript.

The readability of Fig3 4 5 6 is poor.

The author does not present a principal method to grow the depth and increase the width, and the depth and width are critical dimensions for a large-scale model.
 The SOTA dynamic method should include the lastest method(e.g. DWN[1] DFP[2] STR[3] ...)

Besides, this paper only focuses on unstructured sparsity. In fact, the unstructured sparsity is hard to have speed gain emphasized by[4, 5], The authors are suggested conducting experiments on channel/filter pruning[6, 7] or fine-grained structured sparsity [4, 8, 9, 10].

typo: line 358 "onlu".

Overall, the author proposes a framework to reduce **the IMP cost** via transferring "winner ticket" to other neural models. I go through the whole paper. I cannot find any training time report in the experimental evaluation. Therefore, it is not convincing to support the claim with the experiment of this manuscript.

[1] Discovering Neural Wirings

[2] Dynamic Model Pruning with Feedback

[3] soft threshold weight reparameterization for learnable sparsity

[4] Accelerating Sparse Deep Neural Networks

[5] PCNN: pattern-based fine-grained regular pruning towards optimizing CNN accelerators

[6] Neural Pruning via Growing Regularization

[7] Rethinking the Value of Network Pruning

[8] Learning N: M fine-grained structured sparse neural networks from scratch

[9] 1xN Block Pattern for Network Sparsity

[10] PCNN: pattern-based fine-grained regular pruning towards optimizing CNN accelerators

**Time Spent Reviewing:**

7 hours

---

> ### Author Response · Authors · 2021-08-10
> **Response to Reviewer GtXb**
>
> We thank you sincerely for the constructive criticism, and we humbly suggest that many of your questions seem to arise from some confusion on our problem setup, and/or unclarity in our result presentation - for which we apologize. Below, we provide point-to-point clarification and hope those additional explanations could establish our paper’s merit more clearly to you.
>
> ### Response 1: Does our work present “principled growth methods”?
>
> We are sorry we do not understand why you claimed that the current paper “does not present a principal method to grow the depth”. We believe quite the opposite: this paper is aimed to establish the first principled approach to alter the depth of LTH.
>
> Please check: we study various comprehensive ways on how to grow in practice, in Section 3.1; and attempt theoretical rationale exploration, in Section 3.3. As the other two reviewers endorsed, we “perform ablation studies on various choices for the replication/dropping of units” ( Reviewer 9fut), “restrict to adaptation to models of the same family (only differing in depth), and show that the adapted lottery tickets outperform pruning-at-initialization and RigL” (Reviewer 9fut), and “alleviate this disadvantage by transferring the obtained winning ticket to other network structures, showing a good potential for practical application and theoretical exploration” (Reviewer r4hi).
>
> While this current paper is focused on studying the depth aspect for LTH transfer, we are right now studying the transfer of width, by referring to the below paper’s great ideas:
> Ting-Wu Chin, Diana Marculescu, Ari S. Morcos “Width Transfer: On the (In)variance of Width Optimization”, CVPR workshop 2021.
> And we will be happy to share our LTH width transfer progress soon, as another independent work.
>
> If you have any additional questions, we will be happy to address them in the rolling discussion period.
>
> ### Response 2: Compare with SOTA dynamic sparse training methods.
>
> First, we compared with RigL in Section 4.5, which remains as one of the strongest dynamic sparse training methods. Meanwhile, the comparison between one-pass E-LTH and DST is not really “apple-to-apple” fair: please kindly check our Response 4 later for why.
>
> Moreover, we appreciate your suggestions on comparing more DST methods [1-3], and we conducted experiments with DPF [2] during the rebuttal week. The results on CIFAR-10 are shown below. As you can see, E-LTH achieves accuracy results fully on par with DPF, and even better at higher sparsities of ResNet-32 and ResNet-56.
>
> |  Backbone | Original Accuracy (Dense) | Sparsity Level | DPF accuracy (Δ from dense)  | E-LTH accuracy (Δ from dense)  | Mask source for E-LTH |
> |:---------:|:-------------------------:|:--------------:|:----------------------------:|:------------------------------:|:---------------------:|
> | ResNet-20 |           92.48%          |       90%      |             -1.60            |              -1.64             |     From ResNet-14    |
> | ResNet-20 |           92.48%          |       95%      |             -4.47            |              -4.48             |     From ResNet-14    |
> | ResNet-32 |           93.83%          |       90%      |             -1.41            |              -0.96             |     From ResNet-20    |
> | ResNet-32 |           93.83%          |       95%      |             -2.89            |              -2.83             |     From ResNet-20    |
> | ResNet-56 |           94.51%          |       90%      |             -0.56            |              -0.50             |     From ResNet-32    |
> | ResNet-56 |           94.51%          |       95%      |             -1.77            |              -1.66             |     From ResNet-32    |
>
> On the other hand, we also compare the training cost (in FLOPs) of E-LTH and DPF. More specifically, we compare the total number of FLOPs required for E-LTH and DPF to train sparse models with sparsity ratios of 90% and 95% on all five architectures --- ResNet-14, -20, -32, -44, and -56. Using ELTH to achieve this goal, we perform 13 iterations of IMP on ResNet-32 to generate winning tickets with 90% and 95% sparsity ratios and transfer these two tickets to the other four architectures using ETTs. In contrast, DPF needs ten individual runs on the five architectures with two sparsity ratios. We find that E-LTH needs a total of 1.26 billion FLOPs during the above process, while DPF requires 1.49 billion (for each training step with batch size 1). Note that DPF is a dynamic training method and calculates the gradients for all weights (including zero weights), which increases the FLOPs cost; in contrast, E-LTH utilizes static sparsity patterns and thus saves computations during back-propagation by skipping the calculation of the gradients of zero weights. Here, we follow the identical manner as in the RigL paper to estimate the number of FLOPs in back-propagation for dense and E-LTH models.
>
> The above observation implies that E-LTH has great flexibility and efficiency for two reasons. **Firstly**, E-LTH is flexible in terms of the transferability to multiple target architectures. We only need to run IMP once on the source network to get winning tickets and then can transform the winning tickets to as many architectures as we want, as long as they are within the same network family: that is the key ELTH benefit of “reusability” (please see further explained in Response 4). **Secondly**, running IMP naturally generates a series of winning tickets from low to high sparsity ratios. Thanks to this “nested” structure of winning tickets (e.g., the 95% ticket can be found on top of the 90% ticket by pruning an additional 5%, not fully from scratch), the cost of finding tickets on the source model can also be amortized, once we target finding a series of tickets at multiple sparsity ratios. Note that the second point has NOT been leveraged in the above FLOPs computation yet, but can contribute to more “once for all” ticket finding and save training costs in practice.
>
> We will be happy to conduct more experiments with DNW and STR and include their results in the camera-ready version too. Please allow us to emphasize again that the key point of ELTH is reusability via transferring, not one-pass training cost.

---

> > ### Comment · Reviewer_GtXb · 2021-08-23
> > **Thanks for response**
> >
> > Thanks for the detailed response. I am travelling right now and will try to get back on this in a week or so. Apologies for the delay.

---

> > > ### Author Response · Authors · 2021-08-26
> > > **A mild follow-up on our previous response**
> > >
> > > Dear Reviewer GtXb,
> > >
> > > No worries! We hope you have a good time traveling. As the discussion period is approaching its end, we wonder if you could kindly spare some time to look at our response and share some of your thoughts about it. We would sincerely appreciate it!
> > >
> > > Best,
> > >
> > > Paper8053 Authors

---

> > > > ### Comment · Reviewer_GtXb · 2021-08-26
> > > > **Post Rebuttal**
> > > >
> > > > I carefully read the rebuttal, and I raised my rating to 6.

---

> ### Author Response · Authors · 2021-08-10
> **Response to Reviewer GtXb (continued)**
>
> ### Response 3: Channel Pruning?
>
> Thanks for your suggestion. But we humbly argue that channel pruning is not essential to have for our paper, although it remains an interesting future work. By way of context, the authors are experienced in studying both LTH and channel pruning, and are familiar with your listed literature in general (we are happy to cite and discuss them all in the final version). The authors are also knowledgeable about the hardware acceleration effects of various sparsity forms.
>
> Let us start by acknowledging: most (if not all) state-of-the-art LTH and pruning-at-initialization (SNIP, GRASP, SynFlow...) papers use unstructured sparsity and IMP: see Ref. [14] et. al..  One most related exception is perhaps “Drawing Early-Bird Tickets” in ICLR 2020, which is for a different goal of energy-efficient training and unfortunately sacrifices a lot more accuracy than typical LTH definitions.
>
> Therefore, to the authors’ best knowledge, finding a high-quality lottery ticket with structured sparsity remains to be an open and unsolved question, and is beyond the subject of this paper. Since our paper is aimed at improving LTH (as you can tell from the title), it was not developed as a general pruning paper, and we chose to follow the well-accepted standard of this LTH field, by using unstructured sparsity and IMP. This point also seems to be well received by other reviewers.
>
> Besides, while structured sparsity is more friendly to GPU acceleration, the recent development in hardware accelerators shows the promise to turn the unstructured sparsity into practical speedups. For example, in the range of 70%-90% unstructured sparsity, XNNPACK has already shown significant speedups over dense baselines on real smartphone processors:
> Erich Elsen, Marat Dukhan, Trevor Gale, Karen Simonyan "Fast Sparse ConvNets". CVPR 2020.
> Those recent advances are likely to mitigate the gap between the current LTH practice and the hardware practice. We thank you for pointing it out and we are happy to add all the above discussions to the final paper.
>
> ### Response 4: Training time in the experimental evaluation.
>
> **First of all**, our original paper did not emphasize the direct comparison of once-pass training time, for the reason that was already explained in lines 341 - 350 when discussing our comparison with a state-of-the-art DST approach (we copy-paste below for your convenience):
>
> *“A crucial difference between E-LTH and DST is that E-LTH is a “one-for-all” method. For example if we run IMP on ResNet-34 once, we then obtain tickets for ResNet-20, ResNet-44, and more “for free” simultaneously, by applying ETTs... Therefore, any overhead of E-LTH is amortized by transferring to many different-depth architectures: that is conceptually similar to LTH’s value in pre-training [2,3].”*
>
> Therefore, the key merit of E-LTH is to reveal the *reusability* of a found lottery ticket, not a faster one-pass lottery ticket finding. Additional results are also provided in earlier Response 2. This core point seems to be well appreciated by the other two reviewers.
>
> **Second**,  In case you missed them, the same subsection 4.5 has offered training FLOPs number comparison with E-LTH and DST (please read lines 351- 360, and Table 2): the two use similar total FLOPs on training three ResNet models, and E-LTH will become more cost-effective as the found mask is transferred to more networks.
>
> To further address your curiosity, here we report a set of training time numbers of E-LTH experiments in Section 4.5 and compare them with RigL. To train a ResNet-18 on ImageNet following the standard 90-epoch training schedule takes 9.0 hours on 8 Nvidia-V100 GPUs, and 10.0 hours and 10.8 hours for ResNet-26 and ResNet-34 respectively. In Section 4.5, to train sparse models of ResNet-18, -26 and -34 at 73.79% sparsity ratio, E-LTH first runs 7 iterations of IMP on ResNet-26, and then trains the transformed LTs of ResNet-18 and ResNet-34 once, inducing a total (7 * 10.0 + 9.0 + 10.8) = 89.8 hours of training. In contrast, RigL independently trains 3 networks with 5x training steps (default setting in the original RigL paper), inducing a total (9.0 + 10.0 + 10.8) * 5 = 149 hours of training. We will report more training time numbers in the revised version.

---

> ### Author Response · Authors · 2021-08-21
> **Thoughts on the Response Would be Appreciated**
>
> Dear Reviewer GtXb,
>
> We would like to thank you for your great efforts spent reviewing this work. We wonder if our response helps to clarify your concerns.
>
> Especially, the key point we would like to emphasize in our response is the “once-for-all” property and thus reusability of E-LTH, which is the crucial difference that distinguishes E-LTH from DST methods. This property provides greater scalability and efficiency when we transfer LTs to many target architectures of the family. This benefit in efficiency is supported by the FLOPs and training time comparison with DST methods such as RigL and DPF (newly added per your request). Meanwhile, we hope that our argument on “principal growth”, and the unstructured versus structured sparsity, can help to better position our work. We also report more information on training time.
>
> We appreciate it if you could kindly share your thoughts on our response. Thank you!
>
> Best wishes,
>
> Paper8053 Authors

---

### Official Review · Reviewer_r4hi · 2021-07-15

**Rating:** 8
**Confidence:** 4

**Summary:**

This paper focused on exploring the transferability of a subnetwork obtained from the given dense network. Following the Lottery Ticket Hypothesis (LTH), a new hypothesis called Elastic Lottery Ticket Hypothesis (E-LTH) was proposed along with a corresponding validation method––Elastic Ticket Transformations (ETT). Extensive experiments on different benchmarks are provided to prove the effectiveness of ETT.

**Limitations And Societal Impact:**

Suffice.

**Main Review:**

This paper is drafted with a strong motivation resulting in an interesting research topic --- the transferability of a winning ticket. The previous surprising findings in LTH require expensive computational costs to benefit real-world network training. This paper tries to alleviate this disadvantage by transferring the obtained winning ticket to other network structures, showing a good potential for practical application and theoretical exploration.

The experiments are comprehensive. In addition to the comparisons between networks in the same family based on several strong baselines, this work also provides other experimental analyses, such as the comparisons with dynamic sparse training and the analysis of linear mode connectivity.

Some experimental settings are unclear. As mentioned in L257-L258, the sparsity ratio will be different. It is better to provide detailed comparisons and how the “best possible extent” (L259) was made to compensate for the difference.

Some potential missing baselines. I am curious about the extreme situation of both “stretching” and “squeezing.” How about only stretching the last unit to finish transferring from lower to deeper network, or even stretching randomly picked unit.

The value of “squeezing” network: “stretching” network will benefit the winning ticket finding for a larger network based on a smaller one. However, based on the results (e.g., ResNet56 to ResNet32 in Fig. 4), the structural changes in winning tickets will also cause a performance drop for the converse scenario. It is better to elaborate on the potential benefits of this situation.

**Time Spent Reviewing:**

6.5

---

> ### Author Response · Authors · 2021-08-10
> **Response to Reviewer r4hi**
>
> Thank you for your positive response and constructive feedback.
>
> ### Response 1: Clarifying experimental settings and “best possible extent”.
>
> Thank you for pointing this out. Indeed as you can find in all figures, the sparsity ratios (i.e., the values on the x-axis) are not necessarily aligned for all the data points in one iteration of IMP due to the replicating and dropping of blocks (depending on whether the specific blocks have sparsity higher or lower than average). Note that the resultant variations are usually small though. Moreover, those figures provide sufficient information for fair comparison if we look at the global tendency between different curves.
>
> Taking the right sub-figure in Fig. 5 as an example. The data points on the green line usually have higher sparsity ratios compared to their counterparts because we drop the first 4 blocks in each stage of ResNet-56, causing an overall higher sparsity ratio in the transformed tickets. But we can still clearly see that the connected green curve is well below the other curves. Some data points have lower accuracies than their counterparts with much higher sparsity ratios, which correspond to models in higher iterations of IMP. We will clarify the differences in sparsity ratios in the revision.
>
> By saying that “best possible extent” we mean we will select the method that results in closest sparsity ratios to the source winning tickets. For example, when transforming winning tickets of ResNet-32 to ResNet-56, we choose to append or interpolate all 4 blocks in each stage of ResNet-32, as we have verified doing this way will cause the minimal change to the original overall sparsity ratios. We will make this clearer in the revision.
>
> ### Response 2: Missing baselines.
>
> Thank you for suggesting these interesting baselines. We did not include the “stretching last unit” baseline in the submission because we originally intended to show in Figure 3 that replicating more units is better than replicating a single unit and to show by Figure 4 that replicating earlier layers is better. Combining these two observations, we conjecture replicating the last unit is a bad option and thus do not report it. To address your curiosity, we conduct the “replicating the last unit” baseline during the rebuttal period and compare it with “Interpolation - 1234” in Figure 3. The results below show consistent and obvious gaps between “replicating the last unit” baseline from “Interpolation - 1234”. We are happy to include these results in the revised version.
>
> |     IMP Level     |    3   |    4   |    5   |    6   |    7   |    8   |    9   |   10   |
> |:-----------------:|:------:|:------:|:------:|:------:|:------:|:------:|:------:|:------:|
> |     Last Unit     | 92.83% | 92.86% | 92.69% | 92.61% | 92.54% | 92.14% | 91.44% | 91.45% |
> | Interpolate -1234 | 93.16% | 93.22% | 92.96% | 92.92% | 93.06% | 92.78% | 92.75% | 92.70% |
>
> For the “random picking” baseline, we humbly think that it is hard to reliably evaluate this baseline without enough random samplings of replicating strategies, which takes too much time to complete during this limited rebuttal period. However, we are happy to include this baseline in the revised version. We also suggest that the results of many replicating strategies in Figure 3 and 4 could partially imply how the “random picking” baseline would behave.
>
> We also refer you to our responses to Reviewer 9fut, where we include more baselines that also utilize the source winning tickets. We believe these new results further strengthen our work.
>
> ### Response 3: The value of squeezing networks.
>
> Squeezing the winning tickets target potential applications for edge AI, especially when the edge devices are heterogeneous with *different* computational resources available. In this case, it is very useful to adaptively transform the winning tickets of a large network into *different* smaller sizes to fit the various resource budgets, leading to a family of lottery tickets that can fit different performance-resource trade-offs, instead of a single one. Our E-LTH could lead to “once-for-all '' lottery ticket finding, e.g., flexibly obtaining the right lottery ticket for each new target performance-resource trade-off, without reapplying the expensive IMP repeatedly each time. We believe that is a well-desired flexibility towards applying LTH in the edge AI practice, therefore making E-LTH practically significant.

---

### Official Review · Reviewer_9fut · 2021-07-16

**Rating:** 6
**Confidence:** 4

**Summary:**

This paper describes a method for adapting lottery tickets (LT) found in one network architecture to a related architecture of different size. The motivation for this work is to study the connection between LTs of different architectures, and to reduce the cost of performing iterative magnitude pruning (IMP). If LTs can transfer between architectures, then it may be possible to find lottery ticket only once and adapting the size of the model as needed, without having to perform IMP for each model architecture. They restrict to adaptation to models of the same family (only differing in depth), and show that the adapted lottery tickets outperform pruning-at-initialization and RigL.

Networks are viewed at the level of "units", which are layers (+normalization), or residual blocks for ResNets. The adaptation method is to replicate nearby units of a smaller source model for each additional layer, or drop nearby units of a larger source model for each extra layer. They perform ablation studies on various choices for the replication/dropping of units.

**Limitations And Societal Impact:**

Yes

**Main Review:**

Originality:
- This work appears to be very original, as it is the first work I have seen for adapting one lottery ticket to multiple architectures.

Quality:
- My major issue with this work is the claim that lottery tickets (LT) can be stretched or shrunk to different architectures. This is a bold claim and the current experiments leave room for doubt. To claim that the new network is a LT, we should see that 1) reinitialization under the LT mask, 2) LT weights with random mask, and 3) random weights with random mask all perform worse than the LT (while keeping the source blocks the same). However, the "Interpolation 1234 - Random" experiment in Figure 3 shows that it is not significantly different from "Interpolation 1234", raising serious doubts about whether the approach generates "lottery tickets" or just a way to stretch or shrink existing models. No other experiments are conducted to establish that the new network is indeed a LT, which also raises questions.

- It’s unclear how much of the gain is due to simply having access to the LT on the original source model. For pruning-at-initialization baselines, what if you copy the same weights as LTH for existing blocks, and only perform pruning for the new layers using pruning-at-initialization techniques? This would be a more appropriate baseline to compare to. The current experiments disadvantages other baselines by leveraging a prior LT model only for ETTs.

- Are the baselines (unpruned, IMP) initialized in the same way as the elastic LTH? I.e. Weights are repeated for certain blocks? Or is it done with random initializations? If the latter, I would be curious to see what happens when you replicate blocks in the same way at initialization. How much of the benefit of ELTH comes from replicating initial weights (if any)?

- The claims that ETTs may transfer to general pruned solutions seems unfounded (also unclear what "general pruned solutions" mean here, so I may be misreading). To make the claim that LTs derived from different model architectures are connected, and that they stay in the same basin, the authors could look at linear mode connectivity between IMP solution on a target model and the ETT solutions.

Clarity:
- The paper is generally well written. My only comment is that the takeaway of section 4.4 confuses me. I hope the authors can elaborate further on what the implication of their linear mode connectivity result is.

Significance:
- The paper raises a question that would be significant to the community. However, the results so far do not make a compelling case for it. Nonetheless, I would be convinced otherwise if my earlier concerns were sufficiently addressed.


=================================================================
Updates post rebuttal:
The authors have provided additional experiments and clarifications that addressed my earlier concerns. Although the performance difference between ELT and the appropriate baselines suggested in the review is small, it seems consistent, and would be interesting to the community given the general novelty of this work. Assuming that these new results will be added to the paper accordingly, I will increase my score from 4 to 6.

**Time Spent Reviewing:**

6

---

> ### Author Response · Authors · 2021-08-10
> **Response to Reviewer 9fut**
>
> Thank you so much for appreciating our originality, motivation, comprehensive experiments, and writing quality! We have clarified all your concerns below, and we would sincerely appreciate it if you could take another look and let us know if there is any further question.
>
> ### Response 1: Missing baselines to claim that the new network is a LT.
>
> Thank you for suggesting the three baselines. Firstly, the stretching option “Interpolation 1234 - Random '' shown in Figure 3 exactly corresponds to the second baseline you mentioned, i.e., “LT weights with random mask, while keeping the source blocks the same”. Although the performance of this baseline and the “Interpolation 1234” option looks close at some sparsity ratios (esp. smaller sparsity) in Figure 3, “Interpolation 1234” is actually consistently better than “Interpolation 1234 - Random'' at all non-trivial sparsity ratios (>= 50%). The complete set of numbers are shown in the table below. The gaps between them become quite obvious at higher than 90% sparsity ratios (up to 0.55%), which implies that the E-LTH solution does transfer more useful sparsity patterns than just reusing the source blocks. Note that all the results are the average of three independent runs, which we think makes our results reliable.
>
> We also run the experiments of the other two baselines you mentioned during the rebuttal week. The results are summarized in the table below. Here, we name the three baselines as “Random Init”, “Random Mask” and “Random Init+Mask” following the order that you suggest, and show their accuracy drops (-%) than ElasticLTH. Similarly, the gaps between each of them and E-LTH grow as the sparsity rises higher, and become very obvious at high sparsity levels.
>
> | Sparsity Ratio | ElasticLTH | Random Init | Random Mask | Random Init+Mask | Pruning-at-initialization |
> |:--------------:|:----------:|:-----------:|:-----------:|:----------------:|:-------------------------:|
> |       50%      |   93.16%   |    -0.23%   |    -0.30%   |      -0.16%      |           -0.20%          |
> |       60%      |   93.22%   |    -0.21%   |    -0.12%   |      -0.11%      |           -0.31%          |
> |       75%      |   92.92%   |    +0.03%   |    -0.04%   |      -0.15%      |           -0.00%          |
> |       80%      |   93.06%   |    -0.02%   |    -0.25%   |      -0.18%      |           -0.35%          |
> |       84%      |   92.78%   |    -0.12%   |    -0.12%   |      -0.13%      |           -0.09%          |
> |       90%      |   92.70%   |    -0.62%   |    -0.35%   |      -0.39%      |           -0.29%          |
> |       94%      |   91.39%   |    -0.79%   |    -0.42%   |      -0.72%      |           -0.48%          |
> |       95%      |   90.54%   |    -0.66%   |    -0.55%   |      -0.66%      |           -0.27%          |
>
> ### Response 2: New baseline by incorporating pruning-at-initialization methods.
>
> Thank you for the great suggestion! Combining the blocks in the source winning tickets with pruning-at-initialization solutions is an interesting baseline to compare with. During the rebuttal period, we try this baseline exactly as you suggested, and show the results in the table above (in the last column named “Pruning-at-initialization”). We observe similar results as the three baselines in the last question --- ElasticLTH is consistently better than pruning-at-initialization methods, although the source blocks are used in both cases. We hope that these results in the above table can convince you that ElasticLTH does find better tickets than the other baselines and hence can positively change your evaluation of our work.
>
> ### Response 3: How are the baselines initialized?
>
> For all experiments, the baselines (unpruned, IMP, SNIP/GrasP) start from exactly the same initialization as the Elastic LTH (i.e., weights are repeated for certain blocks), to make fair comparisons to E-LTH. For E-LTH, we replicate both the mask and random initialization for replicated blocks. We will make these clearer in the revision.
>
> ### Response 4: Transfer “general pruned solutions”, and the takeaway of Section 4.4.
>
> The observation we make in Section 4.4 is that sparse solutions transferred via ETTs can preserve the linear mode connectivity property of lottery tickets. The main take-away regarding E-LTH from Section 4.4 is: ETTs have better linear mode connectivity when the source and target networks have a smaller difference in depth, in which E-LTH will also work better.
>
> Also regarding the pruning remark: lottery tickets transfer better via ETTs when they stay in the same basin (by comparing tickers from ResNet-20 versus from ResNet-14, to ResNet-32). Since recent work found LTH solutions may stay in the same basin as their corresponding pruning solution [10], we hypothesize that ElasticLTH can also help to preserve the property of the pruning solutions.
>
> This is however just a side remark hypothesizing potential future research, rather than anything attached to this paper’s core claims. We can rephrase our statement in Section 4.4 to exactly explain our implication above, or we can remove it if the reviewers consider this statement not clear nor necessary.

---

> > ### Comment · Reviewer_9fut · 2021-08-23
> > **Confidence intervals on results**
> >
> > I want to thank the authors for a thorough response to my review. The new results and clarifications help alleviate my original concerns. As the accuracy drops are fairly small, I wonder if the authors can provide some sense of a confidence interval for the results (perhaps min/max over the different seeds)?

---

> > > ### Author Response · Authors · 2021-08-23
> > > **More detailed results for a better sense of confidence intervals**
> > >
> > > Dear Reviewer 9fut,
> > >
> > > Thank you for your reply. We are glad that our response helped to alleviate your concerns. For your question on the confidence interval:
> > >
> > > All experiment results in this paper and in our previous response are based on individual runs over 3 different seeds. The random seed decides the initialization, the data order, etc., so it is less meaningful to compare accuracy numbers across different seeds and across methods at the same time. Therefore, we first calculate the accuracy difference between E-LTH and other baselines for each seed individually, and then report the gaps separately in the table below. We can see from the results that
> > >
> > > * In many cases, E-LTH is consistently better than the baselines over different seeds. The gaps are larger at high sparsity levels.
> > > * In other cases, E-LTH usually has slightly lower accuracy (gap < 0.05%)
> > > * There are only rare cases where the accuracy gaps have high variance over different seeds. But E-LTH is still comparable “on average” in those cases.
> > >
> > >
> > > | Sparsity Ratio |        | Random Init |        |        | Random Mask |        |        | Random Init+Mask |        |        | Pruning-at-initialization |        |
> > > |:--------------:|:------:|:-----------:|:------:|:------:|:-----------:|:------:|:------:|:----------------:|:------:|:------:|:-------------------------:|:------:|
> > > |                |  run 1 |    run 2    |  run 3 |  run 1 |    run 2    |  run 3 |  run 1 |       run 2      |  run 3 |  run 1 |           run 2           |  run 3 |
> > > |       50%      | -0.23% |    -0.33%   | -0.12% | -0.12% |    -0.40%   | -0.39% | -0.20% |      -0.18%      | -0.09% | -0.46% |           +0.04%          | -0.18% |
> > > |       60%      | -0.37% |    -0.02%   | -0.25% | -0.22% |    -0.26%   | +0.12% | -0.30% |      -0.04%      | +0.01% | -0.52% |           -0.42%          | +0.01% |
> > > |       75%      | -0.06% |    +0.20%   | -0.04% | -0.07% |    -0.07%   | +0.03% | -0.18% |      -0.09%      | -0.17% | -0.35% |           +0.34%          | +0.01% |
> > > |       80%      | -0.25% |    +0.22%   | -0.02% | -0.32% |    -0.24%   | -0.20% | -0.32% |      -0.19%      | -0.04% | -0.46% |           -0.13%          | -0.47% |
> > > |       84%      | +0.10% |    -0.15%   | -0.30% | +0.04% |    -0.02%   | -0.37% | +0.01% |      -0.03%      | -0.42% | -0.13% |           +0.14%          | -0.27% |
> > > |       90%      | -1.06% |    -0.20%   | -0.60% | -0.29% |    -0.26%   | -0.51% | -0.36% |      -0.34%      | -0.47% | -0.34% |           +0.01%          | -0.54% |
> > > |       94%      | -0.34% |    -0.56%   | -1.48% | -0.37% |    -0.22%   | -0.68% | -0.73% |      -1.09%      | -0.33% | -0.19% |           -0.44%          | -0.82% |
> > > |       95%      | -0.26% |    -0.79%   | -0.93% | -0.65% |    -0.09%   | -0.92% | -0.26% |      -0.79%      | -0.93% | +0.39% |           -0.34%          | -0.85% |
> > >
> > >
> > > Best wishes,
> > >
> > > Paper8053 Authors

---

> > > ### Author Response · Authors · 2021-08-26
> > > **Did we address your follow-up question?**
> > >
> > > Dear Reviewer 9fut,
> > >
> > > We show our sincere appreciation for your reply earlier that helped improve the quality of this manuscript. As the discussion period is approaching its end, we wonder if we have properly addressed your following comments and whether you have other suggestions or concerns.
> > >
> > > We always welcome your comments and suggestions and are happy to address them to improve our work. Thank you!
> > >
> > > Best,
> > >
> > > Paper8053 Authors

---

> ### Author Response · Authors · 2021-08-21
> **Thanks for the Review and Your feedback Would Be Very Welcome**
>
> Dear Reviewer 9fut,
>
> We would like to thank you for your valuable time in reviewing this work, and we really hope to have a further discussion to see if our response solves the concerns.
>
> In particular, we would really appreciate it if you could share your thoughts on our argument that E-LTH indeed finds better new tickets, as supported by the new results we presented in our response. For example, as the new results showed, E-LTH is consistently better than several baselines you suggested, and also better than the alternative of combing the pruning-at-initialization methods. Meanwhile, we hope that our explanation on Section 4.4 also makes sense to you.
>
> We genuinely hope that you could kindly check our response. Your feedback will be much appreciated. Thank you!
>
> Best wishes,
>
> Paper8053 Authors

---

### Decision · Program_Chairs · 2021-09-27

**Decision:**

Accept (Poster)

**Comment:**

This paper investigates whether lottery tickets can transfer across architectures of different depths, following up on prior work which showed transfer across datasets for the same architecture. Reviewers all found the clarity and originality to be strong and the topic is of significant interest. There were some concerns regarding the soundness of the claims presented, but these were resolved through additional experiments performed during the rebuttal period. I would strongly encourage the authors to include these experiments in the final version of the paper. Overall, I think this paper presents a valuable contribution and I recommend acceptance.